# Conformational fingerprinting of allosteric modulators in metabotropic glutamate receptor 2

Brandon Wey-Hung Liauw, Arash Foroutan, Michael R Schamber, Weifeng Lu, Hamid Samareh Afsari*[†], Reza Vafabakhsh*

Department of Molecular Biosciences, Northwestern University, Evanston, United States

**Abstract** Activation of G protein-coupled receptors (GPCRs) is an allosteric process. It involves conformational coupling between the orthosteric ligand binding site and the G protein binding site. Factors that bind at non-cognate ligand binding sites to alter the allosteric activation process are classified as allosteric modulators and represent a promising class of therapeutics with distinct modes of binding and action. For many receptors, how modulation of signaling is represented at the structural level is unclear. Here, we developed fluorescence resonance energy transfer (FRET) sensors to quantify receptor modulation at each of the three structural domains of metabotropic glutamate receptor 2 (mGluR2). We identified the conformational fingerprint for several allosteric modulators in live cells. This approach enabled us to derive a receptor-centric representation of allosteric modulation and to correlate structural modulation to the standard signaling modulation metrics. Single-molecule FRET analysis revealed that a NAM (egative allosteric modulator) increases the occupancy of one of the intermediate states while a positive allosteric modulator increases the occupancy of the active state. Moreover, we found that the effect of allosteric modulators on the receptor dynamics is complex and depend on the orthosteric ligand. Collectively, our findings provide a structural mechanism of allosteric modulation in mGluR2 and suggest possible strategies for design of future modulators.

*For correspondence:
hamid.samareh_afsari@
boehringer-ingelheim.com (HSA);
reza.vafabakhsh@northwestern.
edu (RV)

Present address: †Boehringer
Ingelheim Pharmaceuticals, Inc,
Ridgefield, United States

## Editor's evaluation

The authors advance our understanding of the molecular underpinnings of allostery in GPCRs by showing the effects of allosteric modulators of mGluR2 on receptor conformation at distinct sites in the presence and absence of orthosteric modulators. This is important as drugs and drug candidates acting outside the site where the orthosteric or endogenous ligands bind are harder to identify. This work provides insights into allosteric changes at the level of individual receptors and provides a new path for drug discovery that is of interest to studies of GPCRs in health and disease.

## Introduction

G protein-coupled receptors (GPCRs) are the largest family of membrane receptors in humans and are key drug targets due to their role in nearly all physiological processes (*Dorsam and Gutkind, 2007*; *Thal et al., 2018*). Compounds that bind to the defined, endogenous ligand binding pocket in GPCRs are called orthosteric ligands. Many such orthosteric agonists or antagonists have been developed as successful therapies (*Lindsley et al., 2016*). Despite this success, achieving target specificity in closely related receptors has been a long-standing challenge due to high conservation of the orthosteric

binding site. Moreover, tolerability and safety of orthosteric drugs in therapeutic applications have been difficult to achieve for some GPCRs (*Lindsley et al., 2016*).

Recently, allosteric modulators have emerged as a promising class of therapeutic compounds for fine-tuning physiological response of GPCRs with high receptor specificity and pathway specificity. Allosteric modulators bind to allosteric sites which are structurally distinct from the orthosteric pocket, to indirectly tune the response to the orthosteric ligand (*Foster and Conn, 2017*). Major advances in design, synthesis, and screening of small molecule compounds have produced multiple selective and potent allosteric modulators for many GPCRs (*Lindsley et al., 2016*). In addition, improvements in techniques for measuring GPCR activity have helped reveal the complex pharmacological properties of allosteric modulators (*Christopoulos, 2014*; *Leach and Gregory, 2017*) such as probe and cell-type context dependence (*Sengmany et al., 2019*), biased allosteric agonism, and biased modulation (*Makita et al., 2007*; *Sengmany et al., 2017*). Generally, functional characterization of allosteric modulators is done using assays that quantify changes at specific steps of the signaling cascade, downstream of receptor, such as intracellular $Ca^{2+}$ levels, $IP_1$ accumulation, cellular cAMP levels, ERK1/2 phosphorylation levels, or using energy transfer methods to quantify dissociation of signaling proteins. Collectively, these approaches have provided a pharmacological framework for characterizing and profiling allosteric modulators. However, as functional assays measure the effect of modulators downstream of the receptor, they are unable to provide direct mechanistic insight on allosteric modulation at the receptor level.

Advances in methods for structure determination of membrane proteins have yielded atomic structures of many GPCRs bound to different allosteric modulators and provided insight into different ligand binding modalities and distinct modulator-induced conformations (*Bueno et al., 2020*; *Kruse et al., 2013*; *Liu et al., 2019*; *Seven et al., 2021*; *Shaye et al., 2020*; *Srivastava et al., 2014*). However, despite these advances, for many receptors, structures of only a small subset of receptor-modulator combinations have been determined. Moreover, receptor activation and modulation are dynamic processes, and dynamic information is not achievable by structural representations alone. While progress has been made toward understanding the dynamics of allosteric modulation in class A GPCRs (*Gentry et al., 2015*; *Thal et al., 2018*; *Wootten et al., 2013*), more comprehensive mechanisms, especially for large multi-domain GPCRs, are lacking.

Among all GPCRs, the class C GPCRs are distinct as they are structurally modular, possessing a large extracellular domain and functioning as obligate dimers. Notably, the orthosteric ligand-binding site that is typically found within the 7 transmembrane (7TM) domain bundle in class A GPCRs is in the extracellular Venus flytrap (VFT) domain of class C GPCRs. The VFT domain is linked to the 7TM domain via the cysteine-rich domain (CRD) which is a semi-rigid linker domain. Thus, receptor activation is inherently an allosteric process that involves inter-subunit and inter-domain cooperativity. In the class C family, metabotropic glutamate receptors (mGluRs) are responsible for mediating the slow neuromodulatory effects of glutamate to tune synaptic excitability and transmission (*Niswender and Conn, 2010*; *Pin and Bettler, 2016*), making them promising therapeutic targets for treating a range of neurological and psychiatric disorders (*Conn et al., 2009*; *Foster and Conn, 2017*; *Mantas et al., 2022*). Based on structural (*Doré et al., 2014*; *Du et al., 2021*; *Seven et al., 2021*; *Wu et al., 2014*) and mutagenesis (*Farinha et al., 2015*; *Gregory and Conn, 2015*; *Lundström et al., 2011*) studies, the primary mGluR allosteric binding sites were determined to be located within the 7TM domain bundles. Previous work examining allosteric modulation of mGluR conformational dynamics generally used ensemble methods and was focused on the dimeric rearrangement of either the 7TM domain (*Gutzeit et al., 2019*; *Nasrallah et al., 2021*) or the extracellular ligand-binding domain (*Cao et al., 2021*). While these studies of individual domains provide insights into how allosteric modulators affect mGluR structure and dynamics, they are not conducive for the broader fingerprinting of the modulator effect across multiple domains of the receptor. Specifically, how key pharmacological parameters such as efficacy and potency of different orthosteric and allosteric ligands are manifested structurally at different domains, and how positive and negative allosteric modulators achieve their modulatory effect through modifying the receptor's energy landscape are not known.

Here, we used live-cell fluorescence resonance energy transfer (FRET) and single-molecule FRET (smFRET) imaging with non-perturbing site-specific labeling, to explicitly examine and quantify the effects of orthosteric agonists and allosteric modulators on mGluR2 conformation and dynamics at the three structural domains of the receptor (*Figure 1A*). Comparing live-cell imaging results between

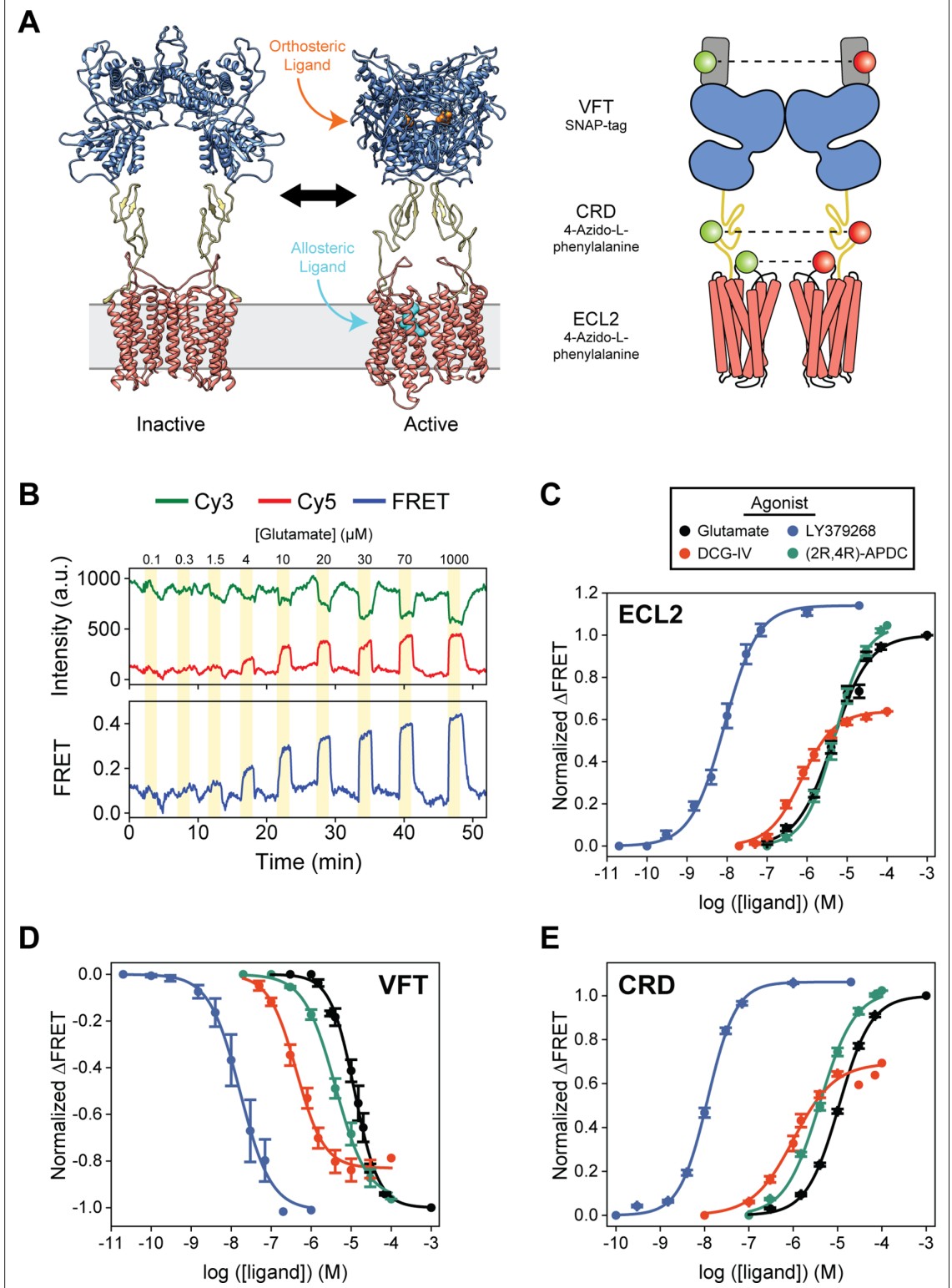

**Figure 1.** Agonist-induced structural change measured at each domain using conformational fluorescence resonance energy transfer (FRET) sensors. (**A**) Full-length cryo-EM structures of inactive (7EPA) and fully active (7E9G) metabotropic glutamate receptor 2 (mGluR2; human) and schematic illustrating fluorophore placement for each inter-domain sensor. (**B**) Representative normalized live-cell FRET trace from glutamate titration experiment on HEK293T cells expressing azi-extracellular loop 2 (azi-ECL2). Data was acquired at 4.5 s time resolution. Dose-response curves from live-cell FRET orthosteric agonist titration experiments using (**C**) azi-ECL2, (**D**) N-terminal SNAP-tag labeled mGluR2 (SNAP-m2), and (**E**) azi-cysteine-rich domain (azi-CRD). Data is acquired from individual cells and normalized to 1 mM glutamate response. Data represents mean ± SEM of responses from individual cells from at

*Figure 1 continued on next page*

*Figure 1 continued*

least three independent experiments. Total number of cells examined, mean half-maximum effective concentration ($EC_{50}$), mean max response, and errors are listed in *Tables 1–2*.

The online version of this article includes the following source data and figure supplement(s) for figure 1:

**Source data 1.** Source data for *Figure 1*.

**Figure supplement 1.** Representative images and fluorescence resonance energy transfer (FRET) traces from live-cell FRET experiments.

**Figure supplement 2.** Quantification of orthosteric agonist efficacy.

**Figure supplement 3.** Orthosteric agonists examined by functional calcium imaging.

the domains, we found that the effect of positive or negative allosteric modulators is represented at every domain of the receptor but to different levels. The effect of modulators on the glutamate efficacy and potency as quantified by the compaction and rearrangement at each receptor domain via the FRET sensors matches with the known functional classification of the compounds. Interestingly, positive allosteric modulators (PAMs) generally increased glutamate efficacy to a greater extent when measured at the CRD and 7TM domains compared to the VFT domain. A similar trend was observed for orthosteric agonists. Our results illustrate that the conformation of the CRD and 7TM domain are more accurate metrics for quantifying ligand efficacy than that of the VFT domain, possibly due to the loose conformational coupling between mGluR2 domains (*Grushevskyi et al., 2019*; *Liauw et al., 2021*). Further examination of the CRD sensor by smFRET revealed that the PAM compound BINA biases more compact intermediate CRD conformations even in the absence of glutamate and reduces the intrinsic CRD dynamics in the presence of glutamate. In contrast, we found that MNI-137, which is a negative allosteric modulator (NAM), blocked receptor activation by impeding CRD progression to the active conformation and preventing glutamate-induced stabilization of the domain. Collectively, the work presented here provides a dynamic receptor-centric model of allosteric modulator effects on mGluR2 conformation and dynamics, as well as mechanisms for positive and negative modulation.

## Results

### CRD and 7TM domain conformation are sensitive measures of mGluR2 activation

According to the general model for mGluR activation, binding of an orthosteric agonist induces a local conformational change that causes global receptor rearrangement to activate the G protein-binding interface 10 nm away, through stabilization of an asymmetric 7TM domain interface (*Seven et al., 2021*). Therefore, activation involves coordinated conformational coupling of the three receptor domains. Structurally, the VFT domain, CRD, and 7TM domain undergo unique dynamics during receptor activation (*Cao et al., 2021*; *Grushevskyi et al., 2019*; *Liauw et al., 2021*). Moreover, how each domain within mGluRs contribute to the overall receptor regulation and activation is now better understood (*Goudet et al., 2004*; *Huang et al., 2011*; *Thibado et al., 2021*). Thus, the three domains can be viewed as modular units that are linked to form a complex and conformationally coupled signaling machine. To gain further insight into mGluR activation and allostery, a better understanding of the dynamics of individual domains and their relation to one another is essential.

Here, we used inter-subunit FRET sensors to measure the dimeric rearrangement of each structural domain within full-length mGluR2 in real-time and in vivo to obtain a more comprehensive picture of receptor activation (*Figure 1A*). Specifically, to study inter-7TM domain conformational change, we created a novel sensor based on an unnatural amino acid (UAA) incorporation strategy (*Huber et al., 2013*; *Liauw et al., 2021*; *Noren et al., 1989*; *Serfling and Coin, 2016*) to site-specifically label extracellular loop 2 (ECL2). We also utilized well established conformational sensors to examine the VFT domain and CRD (*Doumazane et al., 2010*; *Liauw et al., 2021*; *Vafabakhsh et al., 2015*). To generate the inter-7TM domain sensor, we inserted an amber codon between E715 and V716 which, after expression in HEK293T cells, was labeled with 4-azido-L-phenylalanine (hereafter, azi-ECL2). This sensor allowed us to precisely probe conformational changes at ECL2, which have been shown to be essential in coordinating structural transitions between the VFT domain and 7TM domain of not only mGluR2 (*Du et al., 2021*; *Seven et al., 2021*), but other class C GPCRs as well (*Koehl et al., 2019*; *Shen et al., 2021*). We observed a glutamate concentration-dependent increase in

**Table 1.** Live-cell fluorescence resonance energy transfer (FRET) titration experiment data and statistics.

| Sensor | Ligand | N | Mean half-maximum effective concentration (EC$_{50}$) | SEM | Hill slope | Standard error |
|---|---|---|---|---|---|---|
| SNAP-m2 | Glutamate | 9 | 11.9 | 1.5 | −1.44 | 0.08 |
| SNAP-m2 | DCG-IV | 6 | 0.4 | 0.1 | −1.26 | 0.11 |
| SNAP-m2 | LY379268 | 6 | 30.6 | 9.3 | −1.12 | 0.07 |
| SNAP-m2 | (2R,4R)-APDC | 6 | 6.9 | 3.1 | −1.10 | 0.05 |
| SNAP-m2 | Glutamate + 10 µM BINA | 23 | 1.2 | 0.4 | −1.24 | 0.09 |
| SNAP-m2 | Glutamate + 5 µM LY487379 | 4 | 3.8 | 0.9 | −1.43 | 0.11 |
| SNAP-m2 | Glutamate + 0.5 µM JNJ-42153605 | 5 | 4.2 | 1.9 | −0.95 | 0.05 |
| SNAP-m2 | Glutamate + 10 µM MNI-137 | 4 | 17.2 | 2.8 | −1.61 | 0.06 |
| SNAP-m2 | Glutamate + 10 µM Ro 64–5229 | 3 | 19.6 | 2.6 | −1.52 | 0.04 |
| azi-CRD | Glutamate | 26 | 11.6 | 0.5 | 1.19 | 0.03 |
| azi-CRD | DCG-IV | 10 | 1.1 | 0.2 | 0.94 | 0.10 |
| azi-CRD | LY379268 | 20 | 12.1 | 0.5 | 1.36 | 0.05 |
| azi-CRD | (2R,4R)-APDC | 36 | 6.5 | 1.2 | 1.10 | 0.05 |
| azi-CRD | Glutamate + 10 µM BINA | 10 | 1.6 | 0.3 | 1.16 | 0.05 |
| azi-CRD | Glutamate + 5 µM LY487379 | 22 | 4.5 | 0.6 | 0.91 | 0.04 |
| azi-CRD | Glutamate + 0.5 µM JNJ-42153605 | 10 | 4.7 | 1.3 | 0.84 | 0.03 |
| azi-CRD | Glutamate + 10 µM MNI-137 | 27 | 13.8 | 0.7 | 1.10 | 0.04 |
| azi-CRD | Glutamate + 10 µM Ro 64–5229 | 13 | 16.9 | 1.2 | 1.05 | 0.06 |
| azi-ECL2 | Glutamate | 15 | 5.1 | 0.6 | 0.96 | 0.07 |
| azi-ECL2 | DCG-IV | 24 | 0.9 | 0.1 | 1.05 | 0.06 |
| azi-ECL2 | LY379268 | 9 | 10.2 | 2.4 | 1.03 | 0.04 |
| azi-ECL2 | (2R,4R)-APDC | 13 | 6.7 | 1.3 | 1.14 | 0.05 |
| azi-ECL2 | Glutamate + 10 µM BINA | 16 | 2.5 | 0.2 | 1.06 | 0.07 |
| azi-ECL2 | Glutamate + 5 µM LY487379 | 22 | 3.5 | 0.2 | 0.98 | 0.05 |
| azi-ECL2 | Glutamate + 0.5 µM JNJ-42153605 | 17 | 2.2 | 0.1 | 0.97 | 0.06 |
| azi-ECL2 | Glutamate + 10 µM MNI-137 | 8 | 14.4 | 1.7 | 1.32 | 0.06 |
| azi-ECL2 | Glutamate + 10 µM Ro 64–5229 | 5 | 17.4 | 2.4 | 1.07 | 0.09 |

All EC$_{50}$ and errors values are in µM, except for LY379268 (nM).

The online version of this article includes the following source data for table 1:

**Source data 1.** Source data for *Table 1*.

FRET signal in cells expressing azi-ECL2, confirming a general reduction in distance between ECL2s during mGluR2 activation and consistent with structural studies (*Du et al., 2021*; *Seven et al., 2021*; *Figure 1B*, *Figure 1—figure supplement 1A*). This glutamate-dependent increase in ensemble FRET had a half-maximum effective concentration (EC$_{50}$) of 5.1 ± 0.6 µM, consistent with the concentration-dependent activation of GIRK currents (*Vafabakhsh et al., 2015*; *Figure 1C*, *Table 1*). These results validate the sensitivity and accuracy of this new FRET sensor. Next, we measured the concentration-dependent increases in ensemble FRET signals for other orthosteric ligands DCG-IV, LY379268, and (2R,4R)-APDC and measured EC$_{50}$ values of 0.9 ± 0.1 µM, 10.2 ± 2.4 nM, and 6.7 ± 1.3 µM, respectively, in agreement with the published range of EC$_{50}$ values for these compounds (*Doumazane et al., 2013*; *Figure 1C*, *Table 1*, *Figure 1—figure supplement 1*). Importantly, azi-ECL2 accurately reports

**Table 2.** Live-cell fluorescence resonance energy transfer (FRET) max normalization experiment data and statistics.

| Sensor | Ligand | N | Mean max response | SEM |
|--------|--------|---|-------------------|-----|
| SNAP-m2 | Glutamate | - | 1 | - |
| SNAP-m2 | DCG-IV | 25 | 0.79 | 0.01 |
| SNAP-m2 | LY379268 | 23 | 1.01 | 0.01 |
| SNAP-m2 | (2R,4R)-APDC | 14 | 0.96 | 0.01 |
| SNAP-m2 | Glutamate + 10 µM BINA | 7 | 1.02 | 0.01 |
| SNAP-m2 | Glutamate + 5 µM LY487379 | 14 | 1.07 | 0.01 |
| SNAP-m2 | Glutamate + 0.5 µM JNJ-42153605 | 22 | 1.01 | 0.01 |
| SNAP-m2 | Glutamate + 10 µM MNI-137 | 22 | 0.85 | 0.02 |
| SNAP-m2 | Glutamate + 10 µM Ro 64–5229 | 35 | 0.87 | 0.01 |
| azi-CRD | Glutamate | - | 1 | - |
| azi-CRD | DCG-IV | 19 | 0.69 | 0.01 |
| azi-CRD | LY379268 | 25 | 1.06 | 0.02 |
| azi-CRD | (2R,4R)-APDC | 13 | 1.02 | 0.01 |
| azi-CRD | Glutamate + 10 µM BINA | 9 | 1.12 | 0.05 |
| azi-CRD | Glutamate + 5 µM LY487379 | 19 | 1.56 | 0.07 |
| azi-CRD | Glutamate + 0.5 µM JNJ-42153605 | 8 | 1.43 | 0.08 |
| azi-CRD | Glutamate + 10 µM MNI-137 | 18 | 0.86 | 0.02 |
| azi-CRD | Glutamate + 10 µM Ro 64–5229 | 18 | 0.59 | 0.03 |
| azi-ECL2 | Glutamate | - | 1 | - |
| azi-ECL2 | DCG-IV | 25 | 0.64 | 0.02 |
| azi-ECL2 | LY379268 | 22 | 1.14 | 0.04 |
| azi-ECL2 | (2R,4R)-APDC | 56 | 1.05 | 0.01 |
| azi-ECL2 | Glutamate + 10 µM BINA | 14 | 1.42 | 0.07 |
| azi-ECL2 | Glutamate + 5 µM LY487379 | 7 | 1.25 | 0.09 |
| azi-ECL2 | Glutamate + 0.5 µM JNJ-42153605 | 13 | 0.99 | 0.02 |
| azi-ECL2 | Glutamate + 10 µM MNI-137 | 58 | 0.78 | 0.03 |
| azi-ECL2 | Glutamate + 10 µM Ro 64–5229 | 8 | 0.84 | 0.05 |

All max response values are normalized to 1 mM glutamate.

The online version of this article includes the following source data for table 2:

**Source data 1.** Source data for *Table 2*.

that DCG-IV is less efficacious than glutamate, consistent with its characterization as a partial agonist. Likewise, this sensor was able to accurately report on LY379268 and (2R,4R)-APDC which are known to be more efficacious agonists than glutamate (*Figure 1C*, *Table 2*, *Figure 1—figure supplement 2*).

Receptor rearrangement and activation requires local ligand-induced structural change to propagate from the VFT domain through the CRD to the 7TM domain. Thus, we next compared the orthosteric agonist-induced FRET change of azi-ECL2 with that of the VFT domain FRET sensor (N-terminal SNAP-tag labeled mGluR2; hereafter, SNAP-m2) and CRD FRET sensor (labeled via 4-azido-L-phenylalanine insertion at position 548; hereafter, azi-CRD). We found that all three sensors accurately predict the relative efficacy of tested orthosteric ligands (*Figure 1C–E*, *Table 2*, *Figure 1—figure supplement 2A*). Specifically, the three sensors rank the four agonists from most to least efficacious as LY379268 > (2R,4R)-APDC > glutamate > DCG-IV. However, the maximum response by

highly efficacious agonists LY379268 and (2R,4R)-APDC are larger when measured at the CRD and 7TM domain compared to the VFT domain (*Table 2*, *Figure 1—figure supplement 2*). In contrast, maximum response by partial agonist DCG-IV is smaller at the CRD and 7TM domain as compared to measurements at the VFT domain. These findings are consistent with results from our functional calcium imaging assay that utilizes a chimeric G protein (*Conklin et al., 1993*; *Figure 1—figure supplement 3*). For example, DCG-IV shows 79% of glutamate efficacy via the VFT domain FRET sensor, while it shows 69% efficacy via CRD sensor and 64% efficacy via the ECL2 sensor, compared to 69% efficacy via the functional assay. Collectively, the results show that the novel ECL2 sensor accurately report the activation of mGluR2. Moreover, conformation of the CRD and 7TM domain are a more sensitive measure of receptor activation compared to the VFT domain and consistent with the loose coupling between mGluR domains (*Grushevskyi et al., 2019*; *Liauw et al., 2021*).

## Allosteric ligands modulate glutamate potency and efficacy at each structural domain

Establishing general principles to predict physiological outcome of mGluR allosteric modulators has been challenging due to their high context dependence and variability in functional measurements (*Leach and Gregory, 2017*; *Thal et al., 2018*). For example, many mGluR5 PAMs exhibit biased agonism when used in a panel of different functional assays and tested mGluR5 NAMs showed different effects between heterologous and endogenous systems (*Sengmany et al., 2019*; *Sengmany et al., 2017*). To overcome the inherent limitations due to convolution of responses of multiple components in the signaling pathway, we directly quantified the effects of a series of modulators on glutamate-induced rearrangement of mGluR2 using the three FRET sensors described above. This unique approach provides a conformational fingerprint of allosteric modulators, complementing available pharmacological and structural data.

We focused on three PAMs, BINA (*Bonnefous et al., 2005*), LY487379 (*Johnson et al., 2003*), and JNJ-42153605 (*Cid et al., 2012*), and two NAMs, MNI-137 (*Hemstapat et al., 2007*) and Ro 64–5229 (*Kolczewski et al., 1999*). We examined the ability of these compounds to modulate glutamate-induced FRET change of SNAP-m2, azi-CRD, and azi-ECL2 FRET sensors. Specifically, to quantify modulation of glutamate potency ($EC_{50}$), we performed glutamate titrations using each sensor in the presence of a different allosteric modulator. Next, in separate experiments, we derived maximum responses (efficacy) to 1 mM glutamate with and without each of the modulators (*Table 2*, *Figure 2—figure supplements 1–3*). First, glutamate titrations in the presence of all tested PAMs resulted in increased glutamate potency and efficacy at every domain, as measured via FRET (*Figure 2A, C. D, F, G, I*, *Tables 1–2*). Therefore, the positive and negative allosteric modulation, which is defined through signaling assays, are generally manifested consistently at every structural domain of mGluR2. We found that PAMs generally increase glutamate efficacy to a greater extent as probed at the CRD and 7TM domain compared to the VFT domain (*Figure 2J*, *Table 2*). This is similar to the effects we observed for highly efficacious orthosteric agonists LY379268 and (2R,4R)-APDC. Specifically, glutamate efficacy in the presence of 10 μM BINA as reported by azi-CRD and azi-ECL2, and not SNAP-m2, are more consistent with our functional analysis, suggesting that the CRD and 7TM domain are better metrics of ligand efficacy (*Figure 2—figure supplement 4*). Interestingly, JNJ-42153605 showed no change in efficacy as quantified by the FRET signal at ECL2 while it showed changes at VFT domain and CRD (*Figure 2G, I, J*, *Table 2*). The ability of different mGluR2 PAMs to alter glutamate potency and efficacy as probed at each domain and to different degrees suggests that PAMs may utilize distinct mechanisms to achieve allosteric modulation of mGluR2, with each domain distinctly affected by each PAM.

Next, glutamate titration in the presence of NAMs resulted in the overall reduction of glutamate potency and efficacy probed at each of the three domains, as expected for a NAM (*Figure 2B, C, E, F, H, I*, *Tables 1–2*). These results are consistent with our functional calcium imaging assay as well (*Figure 2—figure supplement 4*). Interestingly, at NAM concentration used for FRET imaging (10 μM) we observed robust glutamate-induced conformational change (*Figure 2B, E and H*, *Figure 2—figure supplements 1–3*) but could not detect receptor activation in the presence of glutamate, consistent with previous reports that high concentration of NAMs block mGluR2 signaling (*Hemstapat et al., 2007*; *Kolczewski et al., 1999*). This shows that MNI-137 and Ro 64–5229 can block receptor activation without blocking glutamate-induced conformational change at every domain, even at the 7TM

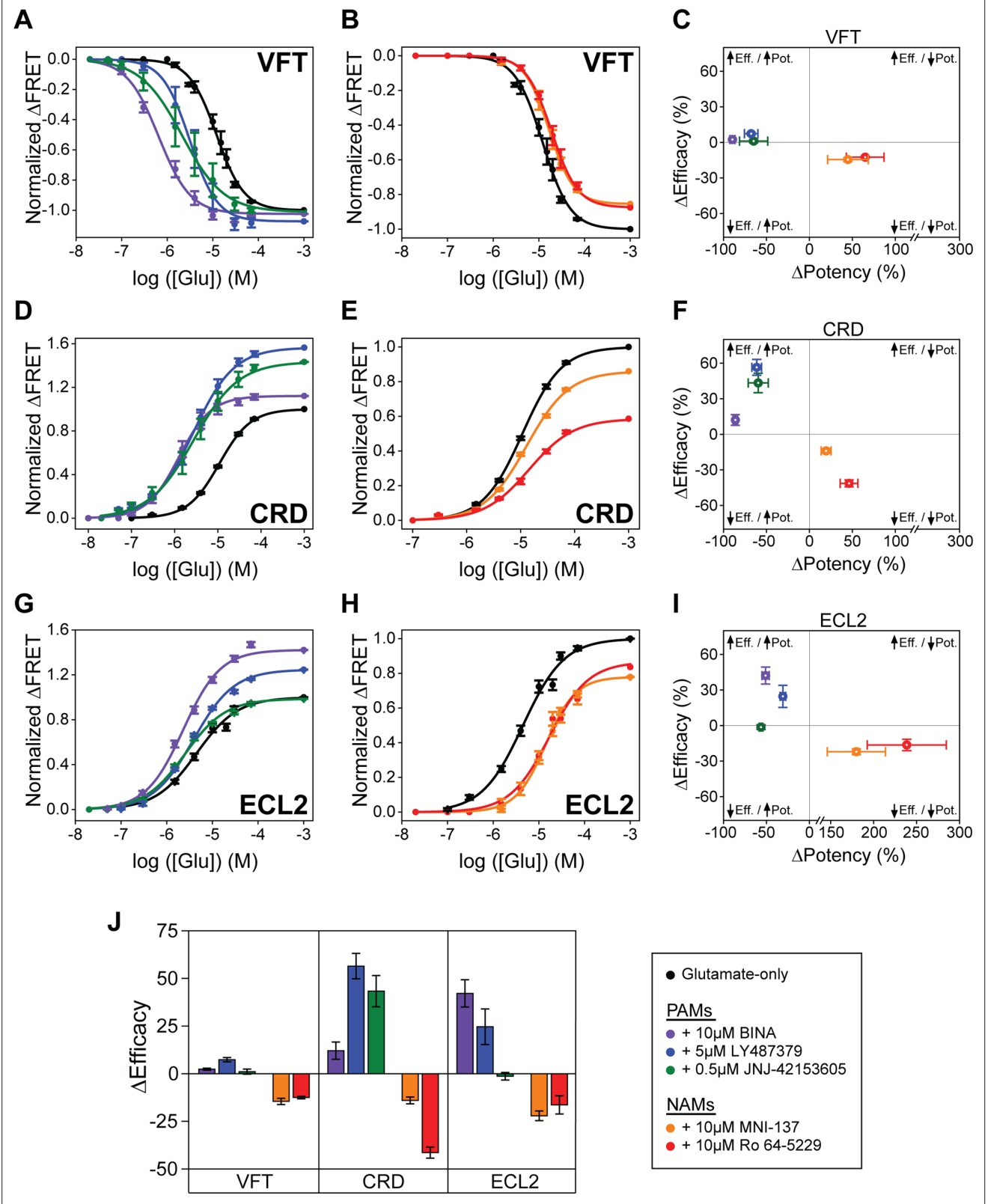

**Figure 2.** Positive and negative allosteric modulation of metabotropic glutamate receptor 2 (mGluR2) structural domains. N-terminal SNAP-tag labeled mGluR2; hereafter (SNAP-m2) glutamate dose-response curves in the presence of (**A**) positive allosteric modulators (PAMs) or (**B**) NAMs. (**C**) Changes in glutamate potency and efficacy for SNAP-m2. The azi-cysteine-rich domain (azi-CRD) glutamate dose-response curves in the presence of (**D**) PAMs or (**E**) NAMs. (**F**) Changes in glutamate potency and efficacy for azi-CRD. The azi-extracellular loop 2 (azi-ECL2) glutamate dose-response curves in

*Figure 2 continued on next page*

*Figure 2 continued*

the presence of (**G**) PAMs or (**H**) NAMs. (**I**) Changes in glutamate potency and efficacy for azi-ECL2. (**J**) Changes in glutamate efficacy in response to PAMs and NAMs as measured by each conformational sensor. ΔPotency defined as (([modulator + glutamate]$_{EC50}$ – [glutamate]$_{EC50}$)/[glutamate]$_{EC50}$) × 100. ΔEfficacy defined as ([1 mM glutamate + modulator] – [1 mM glutamate]) × 100. Data is acquired from individual cells and normalized to 1 mM glutamate response. Data represents mean ± SEM of responses from individual cells from at least three independent experiments. Total number of cells examined for titration and normalization experiments, mean half-maximum effective concentration (EC$_{50}$), mean max response, and errors are listed in *Tables 1–2*.

The online version of this article includes the following source data and figure supplement(s) for figure 2:

**Source data 1.** Source data for *Figure 2*.

**Figure supplement 1.** Max normalization of Δ fluorescence resonance energy transfer (ΔFRET) for N-terminal SNAP-tag labeled metabotropic glutamate receptor 2 (SNAP-m2).

**Figure supplement 2.** Max normalization of Δfluorescence resonance energy transfer (ΔFRET) for azi-cysteine-rich domain (azi-CRD).

**Figure supplement 3.** Max normalization of Δfluorescence resonance energy transfer (ΔFRET) for azi-ECL2.

**Figure supplement 4.** Allosteric modulators examined by functional calcium imaging.

**Figure supplement 5.** Structural representation of allosteric modulator binding pocket.

domain where the NAMs bind. Whether this is due to induction of novel conformational states upon NAM binding or due to interruption in existing conformational changes that precede receptor activation, cannot be addressed using ensemble assays.

Together, the results show that the tested allosteric modulators affect glutamate-induced compaction and activation of mGluR2 in a manner consistent with their functional characterization. Interestingly, while having overlapping binding pockets that share key residues, PAMs and NAMs modulate glutamate-induced conformational change in different ways (*Figure 2—figure supplement 5*). Despite the overall trend for PAMs and NAMs, the general variability in the change of glutamate potency and efficacy between domains in response to individual modulators provides evidence for the existence of multiple pathways to achieve allosteric modulation of mGluR2.

## BINA can function independently of glutamate and stabilizes receptor during activation

Live-cell FRET experiments revealed the general conformational fingerprint of mGluR2 modulators, which are defined as changes in glutamate potency and efficacy as measured by rearrangement of different domains. However, the ensemble method cannot provide mechanistic information such as receptor conformation, state occupancy, and state transitions. For example, whether the modulators stabilize novel states or alter transition rates between existing states is not directly deducible from the ensemble characterization. To overcome this limitation, we performed single-molecule FRET (smFRET) using the CRD FRET sensor. We selected azi-CRD because our live-cell FRET analysis showed that quantification of modulator effects on the CRD was very consistent with our functional results. Moreover, we previously showed azi-CRD to be a sensitive reporter of mGluR2 allosteric modulation via smFRET analysis (*Liauw et al., 2021*).

To perform smFRET imaging, HEK293T cells expressing azi-CRD containing a C-terminal FLAG-tag were labeled using mixture of donor (Cy3) and acceptor (Cy5) fluorophores, then lysed. Cell lysate was then applied to a polyethylene glycol (PEG) passivated coverslip, functionalized with anti-FLAG-tag antibody to immunopurify the receptors (SiMPull) for total internal reflection fluorescence (TIRF) imaging (*Jain et al., 2011*; *Liauw et al., 2021*; *Figure 3A*). In the absence of glutamate, the CRD primarily occupied the inactive state and intermediate state 1, corresponding to open and inactive conformations of the VFT domains or the conformation where an individual VFT domain is closed, respectively (*Liauw et al., 2021*; *Figure 3B and H, Table 3, Figure 3—figure supplement 1A*). Importantly, the receptor showed dynamics between these states. A glutamate scavenging system was added for 0 µM glutamate conditions to ensure no glutamate contamination. Interestingly, in the absence of glutamate and presence of 10 µM BINA, we detected a small increase in FRET, primarily through increased occupancy of intermediate state 2, a conformation in which the 7TM domains are hypothesized to have not formed a stabilizing interaction with one another that is necessary for receptor activation (*Liauw et al., 2021*; *Figure 3E and H, Table 3, Figure 3—figure supplement 2A*). Upon the addition of intermediate (15 µM) and saturating (1 mM) concentrations of glutamate,

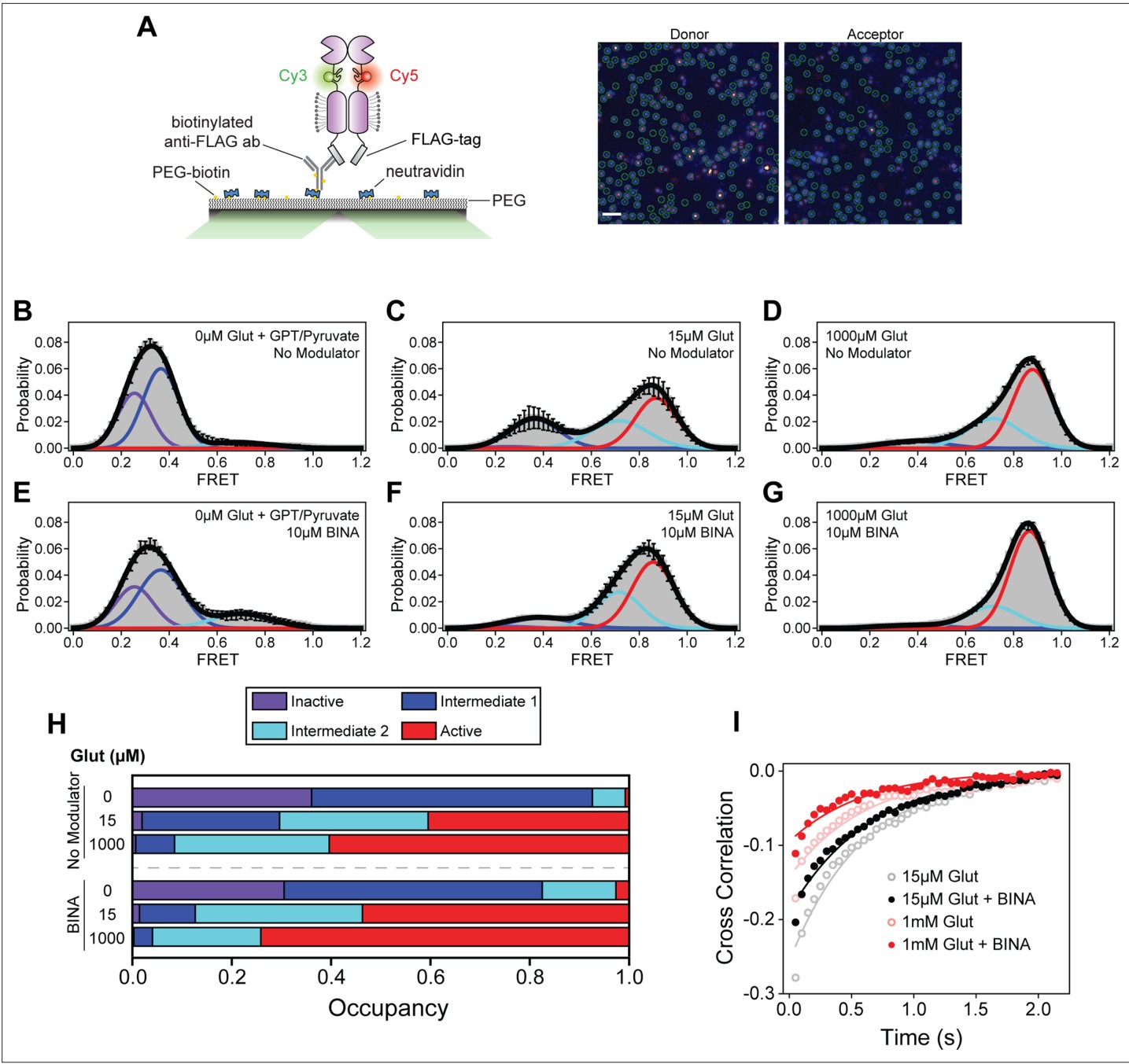

**Figure 3.** Single-molecule fluorescence resonance energy transfer (smFRET) analysis of BINA effects on cysteine-rich domain (CRD) conformational dynamics. (**A**) Schematic of SiMPull assay (left) and representative image of donor and acceptor channels during data acquisition (right). Green circles indicate molecules selected by software for analysis. Scale bar, 3 μm. smFRET population histograms of azi-CRD in the presence of 0 μM, 15 μM, and 1 mM glutamate without (**B–D**) or with (**E–G**) 10 μM BINA. Histograms were fitted (black) to four Gaussian distributions centered around 0.24 (inactive; purple), 0.38 (intermediate 1; blue), 0.70 (intermediate 2; cyan), and 0.87 (active; red) FRET. Error bars represent SEM. Histograms (**B–G**) were generated from 332, 366, 253, 252, 418, and 367 individual particles, respectively. (**H**) Mean occupancy of four conformational states of azi-CRD in varying ligand conditions. Values represent area under each FRET peak from smFRET histogram as a fraction of total area. Mean and SEM values are reported in *Table 3*. (**I**) Mean cross-correlation of donor and acceptor intensities in the presence of intermediate (15 μM) and saturating (1 mM) glutamate with and without 10 μM BINA. Data was acquired at 50 ms time resolution. All data represents mean from three independent experiments.

The online version of this article includes the following source data and figure supplement(s) for figure 3:

**Source data 1.** Source data for *Figure 3*.

**Figure supplement 1.** Representative single-molecule fluorescence resonance energy transfer (smFRET) traces for modulator-free conditions.

**Figure supplement 2.** Representative single-molecule fluorescence resonance energy transfer (smFRET) traces for 10 μM BINA conditions.

**Table 3.** Single-molecule fluorescence resonance energy transfer (smFRET) state occupancy data and statistics.

| Modulator | Glut (µM) | State (#) | Mean occupancy | SEM |
|---|---|---|---|---|
| None | 0 | 1 | 0.36067 | 0.048 |
| None | 0 | 2 | 0.56526 | 0.02692 |
| None | 0 | 3 | 0.06615 | 0.02385 |
| None | 0 | 4 | 0.00792 | 0.00792 |
| None | 15 | 1 | 0.01932 | 0.01049 |
| None | 15 | 2 | 0.27699 | 0.06688 |
| None | 15 | 3 | 0.29899 | 0.01579 |
| None | 15 | 4 | 0.4047 | 0.09147 |
| None | 1000 | 1 | 0.00642 | 0.00292 |
| None | 1000 | 2 | 0.07841 | 0.01209 |
| None | 1000 | 3 | 0.31131 | 0.02404 |
| None | 1000 | 4 | 0.60386 | 0.01743 |
| 10 µM BINA | 0 | 1 | 0.30527 | 0.02468 |
| 10 µM BINA | 0 | 2 | 0.51994 | 0.04492 |
| 10 µM BINA | 0 | 3 | 0.14826 | 0.04699 |
| 10 µM BINA | 0 | 4 | 0.02653 | 0.01748 |
| 10 µM BINA | 15 | 1 | 0.01424 | 0.00761 |
| 10 µM BINA | 15 | 2 | 0.11217 | 0.01526 |
| 10 µM BINA | 15 | 3 | 0.3367 | 0.07918 |
| 10 µM BINA | 15 | 4 | 0.53688 | 0.07621 |
| 10 µM BINA | 1000 | 1 | 0.00296 | 0.00154 |
| 10 µM BINA | 1000 | 2 | 0.03751 | 0.00782 |
| 10 µM BINA | 1000 | 3 | 0.21791 | 0.01663 |
| 10 µM BINA | 1000 | 4 | 0.74162 | 0.01093 |
| 5 µM MNI-137 | 0 | 1 | 0.74861 | 0.02014 |
| 5 µM MNI-137 | 0 | 2 | 0.22198 | 0.01316 |
| 5 µM MNI-137 | 0 | 3 | 0.02038 | 0.01004 |
| 5 µM MNI-137 | 0 | 4 | 0.00903 | 0.00541 |
| 5 µM MNI-137 | 15 | 1 | 0.10387 | 0.02484 |
| 5 µM MNI-137 | 15 | 2 | 0.74937 | 0.01688 |
| 5 µM MNI-137 | 15 | 3 | 0.12724 | 0.01026 |
| 5 µM MNI-137 | 15 | 4 | 0.01952 | 0.00254 |
| 5 µM MNI-137 | 1000 | 1 | 0.00207 | 0.000954 |
| 5 µM MNI-137 | 1000 | 2 | 0.5597 | 0.02561 |
| 5 µM MNI-137 | 1000 | 3 | 0.33098 | 0.03204 |
| 5 µM MNI-137 | 1000 | 4 | 0.10725 | 0.00734 |

The online version of this article includes the following source data for table 3:

**Source data 1.** Source data for *Table 3*.

a concentration-dependent increase in the active state occupancy was observed (*Figure 3C, D and H*, *Table 3*, *Figure 3—figure supplement 1*). The four conformational states and glutamate-dependent increase in FRET agree with previous work (*Liauw et al., 2021*). Specifically, addition of 15 μM glutamate in the presence of 10 μM BINA resulted in a FRET distribution similar to saturating glutamate alone (1 mM), consistent with the effect of PAM on increasing glutamate potency. Finally, 1 mM glutamate plus 10 μM BINA resulted in a further increase in active conformation occupancy, consistent with the effect of PAM on increasing glutamate efficacy (*Figure 3F, G and H*, *Table 3*, *Figure 3—figure supplement 2*). Interestingly, examination of CRD dynamics, as measured by cross-correlation between donor and acceptor intensities, showed that in the presence of intermediate (15 μM) and saturating (1 mM) glutamate concentrations, addition of 10 μM BINA reduced receptor dynamics (*Figure 3I*). Together, these observations suggests that PAMs may increase agonist efficacy by effectively increasing occupancy of the active conformation of the receptor. Moreover, these single-molecule measurements demonstrated that the effect of BINA on mGluR2 conformation and dynamics depends on the presence or absence of glutamate. In the absence of glutamate, BINA increased receptor dynamics and FRET by increasing the occupancy of intermediate state 2 (*Figure 3E and H*, *Table 3*, *Figure 3—figure supplement 2*). While in the presence of intermediate (15 μM) and saturating (1 mM) glutamate, BINA reduced the dynamics of the CRD and increased the occupancy of the active state (*Figure 3F, G, H, I*, *Table 3*, *Figure 3—figure supplement 2*). Interestingly, even in the presence of 1 mM glutamate and BINA, the receptors remained dynamic with the CRDs not fully stabilized in a single conformation.

## MNI-137 prevents CRD progression to the active conformation and glutamate-induced stabilization

Some mGluR2 NAMs that bind at the 7TM domain function as non-competitive antagonists and can prevent glutamate-dependent activation of the receptor (*Hemstapat et al., 2007*). To investigate the molecular mechanism underlying this phenomenon, we next performed smFRET analysis to directly visualize the effect of MNI-137 on the CRD sensor. In the absence of glutamate, 5 μM MNI-137 resulted in a decrease in FRET and increase in occupancy of the inactive conformation of the CRD as compared to unliganded receptor (*Figure 4A and D*, *Table 3*, *Figure 4—figure supplement 1A*). The increase in inactive state occupancy was accompanied by a stabilization of the CRD, demonstrating that MNI-137 reduces intrinsic CRD dynamics in the absence of glutamate, which contrasts with the effects of BINA alone (*Figure 4—figure supplement 2A*). Upon the addition of intermediate (15 μM) and saturating (1 mM) glutamate concentrations and in the presence of 5 μM MNI-137, occupancy of intermediate states 1 and 2 substantially increased with minimal change in the active conformation observed (*Figure 4B–D*, *Table 3*, *Figure 4—figure supplement 1*). To examine which specific state transitions are being hindered by MNI-137, we performed Hidden Markov modeling analysis on the smFRET time traces. Examination of the transition density plots (TDPs) obtained from this analysis showed that at 1 mM glutamate alone the dominant transitions occur between intermediate state 2 and the active conformation for the CRD (*Figure 4E*). This is consistent with the intermediate state 2 being the 'pre-active' conformation (*Liauw et al., 2021*). In contrast, in the presence of both 1 mM glutamate and MNI-137, the CRD primarily transitions between intermediate states 1 and 2, with few transitions to the active state. This suggests that MNI-137 effectively prevents the formation of the stabilizing 7TM domain interaction necessary for mGluR2 activation. Together, these results directly show that MNI-137 prevents receptor activation by blocking the last step toward receptor activation and effectively trapping the receptor in constant transition between the existing intermediate states.

Interestingly, examination of the CRD dynamics by cross-correlation analysis revealed that the effect of MNI-137 on receptor dynamics is dependent on whether glutamate is present or not. In the absence of glutamate, MNI-137 reduced CRD dynamics (*Figure 4—figure supplement 1A*). In contrast, when glutamate and MNI-137 were both present, we observed a glutamate concentration-dependent increase in the CRD dynamics (*Figure 4F*). This effect is the opposite to the effect of BINA, a PAM (*Figure 3I*, *Figure 4—figure supplement 2B*). Thus, in addition to impeding progression of the CRD to the active conformation, MNI-137 also effectively prevents glutamate-induced stabilization of the 7TM domain. Together, these results provide a mechanistic understanding of how MNI-137, a NAM, can block receptor activation. This reduction of CRD stability and blocking of entry into the active conformation also provides insight into why glutamate-induced conformational change can

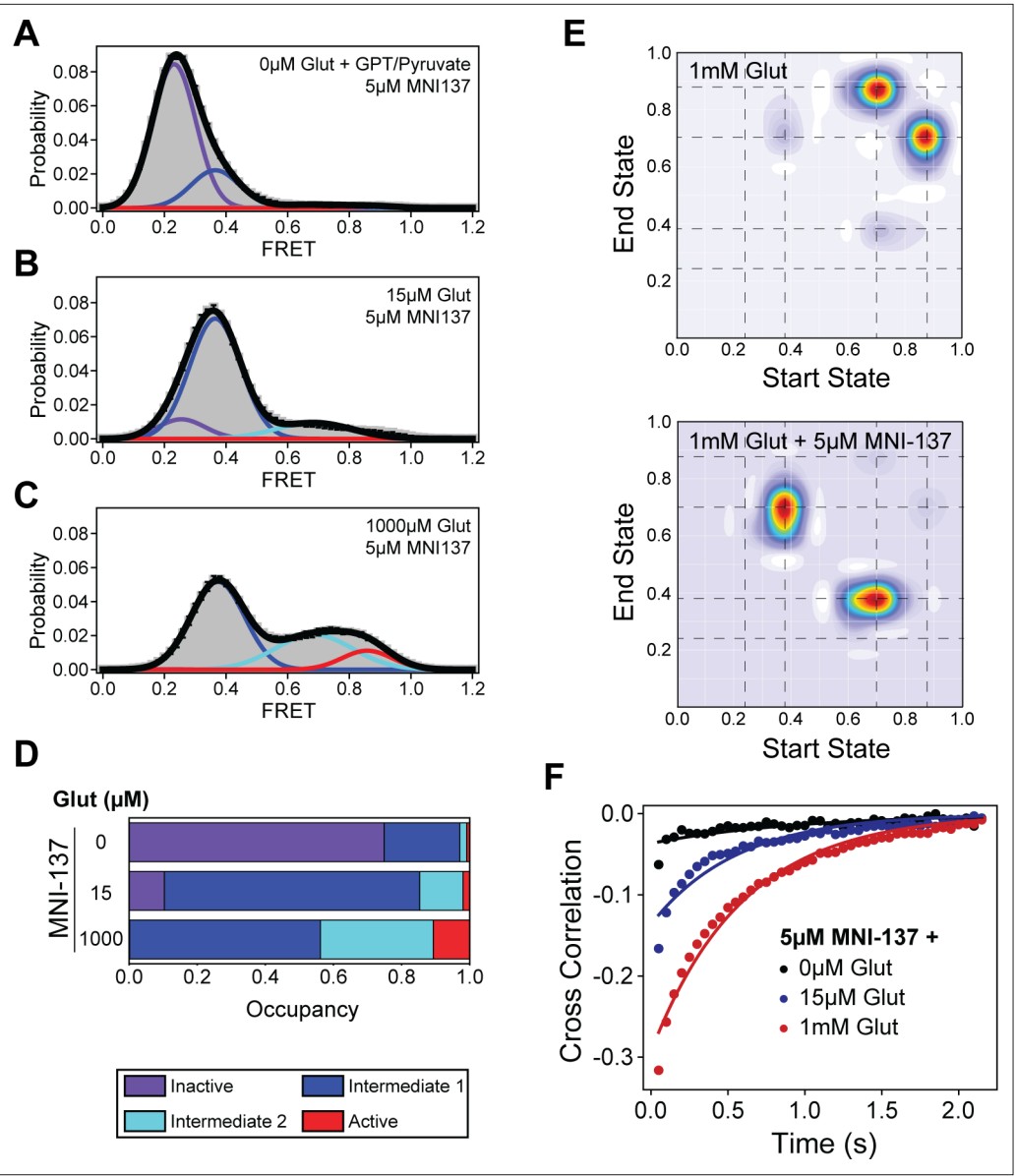

**Figure 4.** Single-molecule fluorescence resonance energy transfer (smFRET) analysis of MNI-137 effects on cysteine-rich domain (CRD) conformational dynamics. (**A–C**) smFRET population histograms of azi-CRD sensor in the presence of 0 µM (372 particles), 15 µM (560 particles), and 1 mM (479 particles) glutamate and 5 µM MNI-137. Histograms were fitted (black) to four Gaussian distributions centered around 0.24 (inactive; purple), 0.38 (intermediate 1; blue), 0.70 (intermediate 2; cyan), and 0.87 (active; red) FRET. Error bars represent SEM. (**D**) Mean occupancy of four conformational states of azi-CRD in varying ligand conditions. Values represent area under each FRET peak from smFRET histogram as a fraction of total area. Mean and SEM values are reported in *Table 3*. (**E**) Transition density plots of azi-CRD at 1 mM glutamate with and without MNI-137. Dashed lines represent four distinct FRET states. (**F**) Mean cross-correlation of donor and acceptor intensities in the presence of 0 µM, 15 µM, and 1 mM glutamate and 5 µM MNI-137. Data was acquired at 50 ms time resolution. Data represents mean from three independent experiments.

The online version of this article includes the following source data and figure supplement(s) for figure 4:

**Source data 1.** Source data for *Figure 4*.

**Figure supplement 1.** Representative single-molecule fluorescence resonance energy transfer (smFRET) traces for 5 µM MNI-137 conditions.

**Figure supplement 2.** Allosteric modulator effects on azi-cysteine-rich domain (azi-CRD) cross correlation.

still be observed, both in live-cell and single-molecule imaging, despite the presence of inhibiting MNI-137 concentrations. Finally, the mechanisms of action for both MNI-137 and BINA highlights the importance of structural dynamics for mGluR activation and modulation.

## Discussion

A fundamental design principle for many receptors is that activation is allosteric in nature. Moreover, ligand 'sensing' and receptor activation is driven by the energy from ligand binding and cellular energy cost in the form of ATP or GTP hydrolysis that occurs after sensing. In GPCRs, activation involves conformational coupling between the ligand binding domain and the G protein binding interface. Recent experiments have shown that GPCRs are dynamic (*Nygaard et al., 2013*) and undergo transition between multiple conformational states, including multiple intermediate states. For class A GPCRs, studies using conformational biosensors based on nuclear magnetic resonance (NMR) spectroscopy (*Huang et al., 2021*), double electron-electron resonance spectroscopy (*Wingler et al., 2019*), smFRET (*Gregorio et al., 2017*), and fluorescent enhancement *Wei et al., 2022* have revealed the importance of conformational dynamics for receptor activation, ligand efficacy, and biased signaling. Specifically, activation of mGluRs involves coordinated movement between three structural domains. In this case, local conformational changes result in major conformational rearrangement that propagate from the ligand binding site to the active site, consistent with the 'domino' model of allosteric signal transduction. Within this framework, allosteric modulators act on sites that are distinct from the orthosteric ligand binding site and affect the function of the receptor. Due to their potential to achieve subtype specificity, allosteric modulators have become a major focus for drug development. Common physiological characterization of GPCR allosteric modulators is often pathway specific and rely on the use of functional assays that quantify the output of the receptor along the signaling cascade. In this work we aimed to develop a receptor-centric view of allosteric modulation by quantifying the relationship between allosteric modulation and protein structural dynamics. Potential sources of heterogeneity arising from differences in post-translational modifications or differences in the local lipid environment, may affect receptor conformation. Therefore, our results represent the average of a heterogeneous population of such receptors. We identified the in vivo conformational fingerprint of multiple allosteric modulators of mGluR2 at three structural domains by using novel non-perturbing FRET sensors. This in vivo approach established a direct connection between the effect of allosteric modulators on receptor conformation at each domain and the physiological metrics of the modulator (i.e. efficacy and potency). Specifically, we found that modulators consistently affect the general trend of glutamate-induced conformational change underlying activation at every structural domain of mGluR2 (*Figure 2*). This result demonstrates the existence of a long-range allosteric pathway along the receptor and over a 10 nm distance. Interestingly, for the same modulator, the degree of conformational change was different among different domains (*Figure 2J*). In fact, we determined that the CRD and 7TM domain conformations are more accurate predictors of ligand efficacy as compared to the VFT domain conformation.

Previous research showed that the activation of mGluR2 is a stepwise process with transitions between four states, including two intermediate states (*Liauw et al., 2021*). Our smFRET analysis with a PAM and NAM showed that allosteric modulators do not induce a new conformational state, within the resolution of smFRET measurements. Instead, they produce their modulatory effect by employing the inherent conformational flexibility of receptors to modify receptor occupancy of the intermediate states. In the case of the PAM, BINA increases the efficacy and potency of glutamate by increasing the transitions from the intermediate state to the active state (*Figure 3*). On the other hand, previous work had shown that the mGluR2 NAM MNI-137 can block receptor signaling. Our analysis provides a mechanism for this observation where MNI-137 blocks entry into the active conformation and increases the transitions into the intermediate states, thereby increasing the occupancy of the intermediate states (*Figure 4*). As a result, the receptor is effectively trapped in the intermediate states. Further studies are necessary to determine the atomic structure of these intermediate states. Interestingly, the regulation of intermediate state occupancy has recently been shown to be a mechanism of allosteric modulation for other classes of GPCRs as well. NMR studies on the μ-opioid receptor (*Kaneko et al., 2022*) and cannabinoid receptor 1 (*Wang et al., 2021*) revealed that PAMs and NAMs regulate receptor function by acting on intermediate conformations in a manner similar to our findings for BINA and MNI-137. Collectively, these results suggest that designing compounds that regulate

intermediate state occupancy is a plausible strategy for the development of allosteric modulators for mGluR2 and other families of GPCRs.

Protein allostery is intimately related to protein dynamics. Our results show that the effect of modulator binding at the 7TM domain on the receptor dynamics probed at the CRD, depends on the orthosteric agonist. In the absence of an orthosteric agonist, NAM stabilize the overall receptor dynamics while PAM increase receptor dynamics (*Figure 4—figure supplement 2A*). On the other hand, in the presence of saturating agonist, the PAM reduced receptor dynamics while the NAM increased receptor dynamics (*Figure 4—figure supplement 2B*). These results further highlight the roles of conformational dynamics in allosteric regulation.

In summary, our study provides a conformational fingerprint of diverse allosteric modulators of mGluR2 at different domains of the receptor. Classically receptors were thought of as two-state switches undergoing transition between on and off states. However, it is now clear that GPCRs' ability to dynamically sample a repertoire of conformations is central to their overall function. Our findings highlight the significance of intermediate states in GPCRs for receptor modulation. Furthermore, our findings suggest that designing compounds that modulate the stability of intermediate states could be a promising direction for developing allosteric drugs. The tools we developed and applied here are not limited to mGluRs and can be extended to the study of other complex multi-domain proteins.

## Materials and methods

**Key resources table**

| Reagent type (species) or resource | Designation | Source or reference | Identifiers | Additional information |
|---|---|---|---|---|
| Cell line (*Homo sapiens*) | HEK 293T | Sigma Aldrich | Cat # 12022001 | |
| Transfected construct (*Mus musculus*) | SNAP-m2 | *Liauw et al., 2021* | | |
| Transfected construct (*Mus musculus*) | SNAP-m2 (no-FLAG) | *Liauw et al., 2021* (modified) | | |
| Transfected construct (*Mus musculus*) | azi-CRD | *Liauw et al., 2021* | | |
| Transfected construct (*Mus musculus*) | azi-ECL2 | Genscript (modified) | ORF clone: OMu19627D | |
| Transfected construct (*Homo sapiens*) | pIRE4-Azi | Addgene | Plasmid # 105,829 | |
| Transfected construct (*Mus musculus*) | Gqo5 | Addgene (modified) | Plasmid # 24,500 | |
| Chemical compound, drug | Glutamate | Sigma Aldrich | Cat # 6106-04-3 | |
| Chemical compound, drug | LY379268 | Tocris | Cat # 2,453 | |
| Chemical compound, drug | DCG-IV | Tocris | Cat # 0975 | |
| Chemical compound, drug | (2R,4R)-APDC | Tocris | Cat # 1,208 | |
| Chemical compound, drug | LY487379 | Tocris | Cat # 3,283 | |
| Chemical compound, drug | BINA | Tocris | Cat # 4,048 | |
| Chemical compound, drug | JNJ-42153605 | Cayman Chemical | 21,984 | |
| Chemical compound, drug | Ro 64–5229 | Tocris | Cat # 2,913 | |
| Chemical compound, drug | MNI-137 | Tocris | Cat # 4,388 | |
| Chemical compound, drug | SNAP-Surface Alexa Fluor 549 | New England Biolabs | S9112S | |
| Chemical compound, drug | SNAP-Surface Alexa Fluor 647 | New England Biolabs | S9136S | |
| Chemical compound, drug | Oregon Green 488 BAPTA-1, AM | Thermo Fisher Scientific | O6807 | |
| Chemical compound, drug | Cy3 Alkyne | Click Chemistry Tools | TA117-5 | |

*Continued on next page*

Continued

| Reagent type (species) or resource | Designation | Source or reference | Identifiers | Additional information |
|---|---|---|---|---|
| Chemical compound, drug | Cy5 Alkyne | Click Chemistry Tools | TA116-5 | |
| Chemical compound, drug | 4-azido-L-phenylalanine | Chem-Impex International | Cat # 06162 | |
| Chemical compound, drug | Aminoguanidine (hydrochloride) | Cayman Chemical | 81,530 | |
| Chemical compound, drug | BTTES | Click Chemistry Tools | 1237–500 | |
| Chemical compound, drug | Copper (II) sulfate | Sigma Aldrich | Cat # 451657–10 G | |
| Chemical compound, drug | (+)-Sodium L-Ascorbate | Sigma Aldrich | Cat # 11140–250 G | |
| Chemical compound, drug | Glutamic-Pyruvic Transaminase | Sigma Aldrich | Cat # G8255-200UN | |
| Chemical compound, drug | Sodium Pyruvate | Gibco | 11360–070 | |
| Chemical compound, drug | DMEM | Corning | 10–013-CV | |
| Chemical compound, drug | Defined Fetal Bovine Serum | Thermo Fisher Scientific | SH30070.03 | |
| Chemical compound, drug | Penicillin-Streptomycin | Gibco | 15140–122 | |
| Chemical compound, drug | Lipofectamine 3000 Transfection Reagent | Thermo Fisher Scientific | L3000015 | |
| Chemical compound, drug | Poly-L-lysine hydrobromide | Sigma Aldrich | Cat # P2636 | |
| Chemical compound, drug | FLAG-tag antibody | Genscript | A01429 | |
| Software, algorithm | smCamera (Version 1.0) | http://ha.med.jhmi.edu/resources/ | | |
| Software, algorithm | ImageJ (Version 1.52 p) | http://imagej.nih.gov/ij/ | RRID:SCR_003070 | |
| Software, algorithm | OriginPro (2020b) | https://www.originlab.com/ | RRID:SCR_014212 | |
| Software, algorithm | Adobe Illustrator (2022) | https://www.adobe.com/ | RRID:SCR_010279 | |

## Molecular cloning

The C-terminal FLAG-tagged mouse mGluR2 construct in pcDNA3.1(+) expression vector was purchased from GenScript (ORF clone: OMu19627D) and verified by sequencing (ACGT Inc). Full length mGluR2 construct with an amber codon (TAG) mutation of amino acid A548 (azi-CRD) or N-terminal SNAP-tag (SNAP-mGluR2) were generated as previously reported (*Liauw et al., 2021*). The insertion of an amber codon (TAG) between E715 and V716 in mGluR2 (azi-ECL2) was performed using the QuikChange site-directed mutagenesis kit (Agilent). SNAP-mGluR2 constructs used for calcium imaging had C-terminal FLAG-tag removed by PCR-based deletion using phosphorylated primers. All plasmids were sequence verified (ACGT Inc). DNA restriction enzymes, DNA polymerase and DNA ligase were from New England Biolabs. Plasmid preparation kits were purchased from Macherey-Nagel.

## Cell culture

HEK293T cells (Sigma) were authenticated (ATCC) and tested for mycoplasma contamination (Lonza). HEK293T cells were maintained in DMEM (Corning) supplemented with 10% (v/v) fetal bovine serum (Fisher Scientific), 100 unit/mL penicillin-streptomycin (Gibco) and 15 mM HEPES (pH = 7.4, Gibco) at 37°C and 5% $CO_2$. The cells were passaged with 0.05% trypsin-EDTA (Gibco). For UAA-containing protein expression, the growth media was supplemented with 0.6 mM 4-azido-L-phenylalanine (Chem-Impex International). All media was filtered by 0.2 µM PES filter (Fisher Scientific).

## Transfection and protein expression

About 24 hr before transfection, HEK293T cells were cultured on poly-L-lysine-coated 18 mm glass coverslips (VWR). For SNAP-mGluR2 used in FRET experiments, media was refreshed with standard growth media and transfected using Lipofectamine 3000 (Fisher Scientific) (total plasmid: 1 µg/18 mm coverslip). Growth media was refreshed after 24 hr and cells were grown for an additional 24 hr.

For UAA-containing protein expression, 1 hr before transfection, media was changed to the growth media supplemented with 0.6 mM 4-azido-L-phenylalanine. mGluR2 plasmids with an amber codon (azi-CRD or azi-ECL2) and pIRE4-Azi plasmid (pIRE4-Azi was a gift from Irene Coin, Addgene plasmid # 105829) were co-transfected (1:1 w/w) into cells using Lipofectamine 3000 (Fisher Scientific) (total plasmid: 2 µg/18 mm coverslip). The growth media containing 0.6 mM 4-azido-L-phenylalanine was refreshed after 24 hr and cells were grown for an additional 24 hr. On the day of the experiment, 30 min before labeling, supplemented growth media was removed and cells were washed by extracellular buffer solution containing (in mM): 128 NaCl, 2 KCl, 2.5 $CaCl_2$, 1.2 $MgCl_2$, 10 sucrose, 10 HEPES, pH = 7.4 and were kept in growth medium without 4-azido-L-phenylalanine.

For calcium imaging experiments, media was refreshed with standard growth media and cells were co-transfected with SNAP-mGluR2 (no FLAG-tag) and chimeric G protein (Gqo5, Addgene plasmid #24500) (1:2 w/w) using Lipofectamine 3000 (Fisher Scientific) (total plasmid: 1.5 µg/18 mm coverslip). For calcium imaging using UAA-containing proteins (azi-CRD or azi-ECL2), we followed the transfection and growth protocol described above and included an additional 1 µg of chimeric G protein (Gqo5). Growth media was refreshed after 24 hr, and cells were grown for an additional 24 hr. Before the addition of labeling solutions, cells were washed with extracellular buffer solution.

## SNAP-tag labeling for FRET measurements

SNAP-tag labeling of SNAP-mGluR2 was done by incubating cells with 2 µM of SNAP-Surface Alexa Fluor 549 (NEB) and 2 µM of SNAP-Surface Alexa Fluor 647 (NEB) in extracellular buffer for 30 min at 37°C. After labelling, cells were washed by extracellular buffer solution to remove excess dye.

## UAA labeling by azide-alkyne click chemistry

The UAA labeling by azide-alkyne click chemistry was performed as previously reported (*Liauw et al., 2021*). Stock solutions were made as follows: Cy3 and Cy5 alkyne dyes (Click Chemistry Tools) 10 mM in DMSO, BTTES (Click Chemistry Tools) 50 mM, copper (II) sulfate (Sigma) 20 mM, aminoguanidine (Cayman Chemical) 100 mM, and (+)-sodium L-ascorbate (Sigma) 100 mM in ultrapure distilled water (Invitrogen). In 656 µL of extracellular buffer solution, Cy3 and Cy5 alkyne dyes were mixed to a final concentration of 18 µM for each dye. To this mixture, a fresh pre-mixed solution of copper (II) sulfate and BTTES (1:5 molar ratio) was added at the final concentration of 150 µM and 750 µM, respectively. Next, aminoguanidine was added to the final concentration of 1.25 mM. Lastly, (+)-sodium L-ascorbate was added to the mixture to a final concentration of 2.5 mM. Total labeling volume was 0.7 mL. The labeling mixture was incubated at 4°C for 8 min, followed by a 2 min incubation at room temperature before addition to cells. Cells were washed with extracellular buffer solution prior to addition of labeling mixture. During labeling, cells were kept in the dark at 37°C and 5% $CO_2$. After 10 min, L-glutamate (Sigma) was added to the cells to a final concentration of 0.5 mM and cells were incubated for an additional 5 min. After labeling, cells were washed by the extracellular buffer solution to remove excess dye.

## Labeling for calcium imaging

Cells used for calcium imaging experiments were labeled using 1 µM SNAP-Surface Alexa Fluor 647 (NEB) and 4 µM Oregon Green 488 BAPTA-1 (Fisher Scientific) in extracellular buffer for 30 min at 37°C. For cells expressing UAA-containing proteins, we labeled the cells with 4 µM Oregon Green 488 BAPTA-1. After labeling, cells were washed by extracellular buffer solution to remove excess dye.

## Live-cell FRET measurements

The microscope and flow system setup used were as previously reported (*Liauw et al., 2021*). After labeling, coverslip was assembled in the flow chamber (Innova Plex) and attached to a gravity flow control system (ALA Scientific Instruments). Extracellular buffer solution was used as imaging buffer and applied at the rate of 5 mL min⁻¹. Labeled cells were imaged on a home-built microscope equipped with a × 20 objective (Olympus, oil-immersion) and using an excitation filter set with a quad-edge dichroic mirror (Di03-R405/488/532/635, Semrock) and a long-pass filter (ET542lp, Chroma). All data were recorded at 4.5 s time resolution for UAA containing constructs and 4 s for SNAP-tag containing constructs. All experiments were performed at room temperature. Donor fluorophores

were excited with a 532 nm laser (RPMC Lasers) and emissions from donor and acceptor fluorophores were simultaneously recorded.

Analysis of live-cell FRET data was performed using smCamera (http://ha.med.jhmi.edu/resources/), ImageJ (http://imagej.nih.gov/ij/), and OriginPro (OriginLab). Movies were corrected for bleed-through of the donor signal into the acceptor channel. Donor bleed-through correction was done by measuring signals from 50 ROIs of Cy3 labeled cells in both the donor and acceptor channels and was calculated to be 8.8%. ROIs used for analysis included the whole cell membrane for individual cells. Apparent FRET efficiency was calculated as FRET = $(I_A - 0.088 \times I_D)/(I_D + (I_A - 0.088 \times I_D))$, where $I_D$ and $I_A$ are the donor and acceptor intensity after buffer-only background subtraction. ΔFRET was calculated as the difference between FRET signal during treatment condition and FRET signal before treatment. In each case, the fluorescence was averaged over 6 datapoints once the signal was stable. Dose-response equation $y(x) = A1 + \frac{A_2 - A_1}{1 + 10^{(\log x_0 - x)P}}$ was used for fitting FRET response to calculate $EC_{50}$ values, where $A1$ is the lower asymptote, $A2$ is the upper asymptote, $P$ is the Hill slope, and $x_0$ is the $EC_{50}$. Maximal responses were normalized to 1 mM glutamate response. All data is from at least three independent biological replicates.

As analysis was limited to relative FRET changes between drug treatments rather than absolute FRET values, no further corrections, aside from the 8.8% bleed-through subtraction, were applied. A small artifact in Cy3 signal (decrease in fluorescence) was observed in response to modulator application for donor-only labeled cells. However, this response showed the same relative amplitude and kinetics as FRET responses and were similar among all modulators tested, thus, was not corrected for. All analyzed FRET changes were verified showing anti-correlated behavior. Furthermore, analysis of acceptor signal in response to different modulator treatment qualitatively recapitulated results of FRET data.

## Calcium imaging

After labeling, sample was assembled in the flow chamber (Innova Plex) and attached to the flow control system (ALA Scientific Instruments) in an identical manner to live-cell FRET experiments. Labeled cells were imaged using an inverted confocal microscope (Zeiss, LSM-800) with a × 40 oil-immersion objective (Plan-Apochromat × 40/1.3oil DIC (UV) VIS-IR M27). Sample was illuminated using a 488 nm laser and fluorescence from Oregon Green 488 nm was measured by a GaAsP-PMT detector with detection wavelengths set to 410–617 nm. For cells expressing SNAP-mGluR2 (no FLAG-tag), samples were excited using the 488 nm laser and a 640 nm laser simultaneously, and Cy5 fluorescence was measured with detection wavelengths set to 648–700 nm. All calcium imaging data were recorded at 3 s time resolution and at room temperature.

Analysis of functional calcium imaging data was performed using ImageJ (http://imagej.nih.gov/ij/) and OriginPro (OriginLab). All cells showing agonist-induced calcium response were selected for initial analysis, with those showing significant drift or photobleaching being omitted from downstream analysis. Fluorescence signal was measured for individual cells from a given movie, normalized from 0 to 1, and averaged. Changes in calcium signal were calculated from these averaged responses as the difference between max response during treatment and response before treatment. Baseline signal intensity was the average over 6 datapoints prior to treatment application. Dose-response equation $y(x) = A1 + \frac{A_2 - A_1}{1 + 10^{(\log x_0 - x)P}}$ was used for fitting calcium response to calculate $EC_{50}$ values, where $A1$ is the lower asymptote, $A2$ is the upper asymptote, $P$ is the Hill slope, and $x_0$ is the $EC_{50}$. Maximal responses were calculated as a fraction of 10 μM ionomycin-induced response, then normalized to 1 mM glutamate response. Direct activation of mGluR2 and subsequent intracellular calcium flux caused by the positive allosteric modulators LY487379 and JNJ-42153605 precluded analysis of the compounds ability to affect glutamate potency and efficacy. All data are from three independent biological replicates.

## smFRET measurements

Single-molecule experiments were conducted using custom flow cells prepared from glass coverslips (VWR) and slides (Fisher Scientific) passivated with mPEG (Laysan Bio) and 1% (w/w) biotin-PEG to prevent unspecific protein adsorption, as previously described (*Jain et al., 2011*; *Vafabakhsh et al., 2015*). Prior to experiments, flow cells were functionalized with FLAG-tag antibody. This was achieved by first incubating flow cells with 500 nM NeutrAvidin (Fisher Scientific) for 2 min followed by 20 μM

biotinylated FLAG-tag antibody (A01429, GenScript) for 30 min. Unbound NeutrAvidin and bioti-nylated FLAG-tag antibody were removed by washing between each incubation step. Washes and protein dilutions were done using T50 buffer (50 mM NaCl, 10 mM Tris, and pH 7.4).

After labeling, cells were recovered from an 18 mm poly-L-lysine coverslip by incubating with $Ca^{2+}$-free DPBS followed by a gentle pipetting. Cells were then pelleted by a 4000 $g$ centrifuga-tion at 4°C for 10 min. The supernatant was removed and cells were resuspended in 100 µL lysis buffer consisting of 200 mM NaCl, 50 mM HEPES, 1 mM EDTA, protease inhibitor tablet (Fisher Scientific), and 0.1 w/v% LMNG-CHS (10:1, Anatrace), pH 7.4. Cells were allowed to lyse with gentle mixing at 4°C for 1 hr. Cell lysate was then centrifuged for 20 min at 20,000 $g$ and 4°C. The superna-tant was collected and immediately diluted 10-fold with dilution buffer consisting of 200 mM NaCl, 50 mM HEPES, 1 mM EDTA, protease inhibitor tablet, and 0.0004 w/v% GDN (Anatrace), pH 7.4. The diluted sample was then added to the flow chamber to achieve sparse surface immobilization of labeled receptors by their C-terminal FLAG-tag. After optimal receptor coverage was achieved, flow chamber was washed extensively (>20 × chamber volume) to remove unbound proteins and excess detergent with wash buffer consisting of 200 mM NaCl, 50 mM HEPES, 0.005 w/v% LMNG-CHS (10:1, Anatrace), and 0.0004 w/v% GDN, pH 7.4. Finally, labeled receptors were imaged in imaging buffer consisting of (in mM) 128 NaCl, 2 KCl, 2.5 $CaCl_2$, 1.2 $MgCl_2$, 40 HEPES, 4 Trolox, 0.005 w/v% LMNG-CHS (10:1), 0.0004 w/v% GDN, and an oxygen scavenging system consisting of protocatechuic acid (Sigma) and 1.6 U/mL bacterial protocatechuate 3,4-dioxygenase (rPCO) (Oriental Yeast Co.), pH 7.35. For glutamate-free conditions, imaging buffer contained 2 U/mL glutamic-pyruvic transami-nase (Sigma) and 2 mM sodium pyruvate (Gibco) and was incubated at 37°C for 10 min. All reagents were prepared from ultrapure-grade chemicals (purity >99.99%) and were purchased from Sigma. All buffers were made using ultrapure distilled water (Invitrogen). Samples were imaged with a 100 × objective (Olympus, 1.49 NA, Oil-immersion) on a custom-built microscope with 50ms time resolution unless stated otherwise. 532 nm and 638 nm lasers (RPMC Lasers) were used for donor and acceptor excitation, respectively.

## smFRET data analysis

Analysis of single-molecule fluorescence data was performed using smCamera (http://ha.med. jhmi.edu/resources/), custom MATLAB (MathWorks) scripts, and OriginPro (OriginLab). Particle selection and generation of raw FRET traces were done automatically within the smCamera soft-ware. For the selection, particles that showed acceptor signal upon donor excitation, with acceptor brightness greater than 10% above background and had a Gaussian intensity profile, were automat-ically selected and donor and acceptor intensities were measured over all frames. Out of this pool, particles that showed a single donor and a single acceptor bleaching step during the acquisition time, stable total intensity ($I_D + I_A$), anti-correlated donor and acceptor intensity behavior without blinking events, and lasted for more than 4 s were manually selected for further analysis (~20%–30% of total molecules per movie). All data was analyzed by three individuals independently and the results were compared and showed to be identical. In addition, a subset of data was blindly analyzed to ensure no bias in analysis. Apparent FRET efficiency was calculated as ($I_A − 0.088 \times I_D$)/($I_D + (I_A − 0.088 \times I_D)$), where $I_D$ and $I_A$ are raw donor and acceptor intensities, respectively. Experi-ments were conducted on three independent biological replicates, to ensure reproducibility of the results. Population smFRET histograms were generated by compiling at least 250 total FRET traces of individual molecules from all replicates. Before compiling traces, FRET histograms of individual molecules were normalized to 1 to ensure that each trace contributes equally, regardless of trace length. Error bars on histograms represent the standard error of data from three independent biological replicates.

Peak fitting analysis on population smFRET histograms was performed with OriginPro and used four Gaussian distributions as $y(x) = \sum_{i=1}^{4} \frac{A_i}{w_i \sqrt{\frac{\pi}{2}}} e^{-2 \frac{(x-x_{ci})^2}{w_i^2}}$ ,where $A$ is the peak area, $w$ is the peak width, and $xc$ is the peak center. Peak areas were constrained to $A > 0$. Peak widths were constrained to $0.1 \leq w \leq 0.25$. Peak centers were constrained to ±0.015 of mean FRET efficiency of each confor-mational state. The mean FRET efficiencies of the inactive state, intermediate state 1, intermediate state 2, and the active state were assigned to 0.24, 0.38, 0.70, and 0.87, respectively, based on the most common FRET states observed in TDPs. This analysis is described in further detail below. State

occupancy probability was calculated as area of specified peak relative to total area, which is defined as the sum of all four individual peak areas.

Raw donor, acceptor, and FRET traces were idealized with a hidden Markov model (HMM) using vbFRET software (*Bronson et al., 2009*; *Zhang et al., 2018*). Transitions, defined as ΔFRET >0.1, were extracted from idealized fits and used to generate TDPs. In situations where the HMM fit does not converge to the data (e.g. due to long fluorophore blinking events or large non-anticorrelated intensity fluctuations), traces were omitted from downstream analysis.

The cross-correlation (CC) of donor and acceptor intensity traces at time $\tau$ is defined as

$$CC\left(\tau\right) = \frac{\delta I_D\left(t\right)\delta I_A\left(t+\tau\right)}{<I_D\left(t\right)>+<I_A\left(t\right)>},$$

where $\delta I_D\left(t\right) = I_D\left(t\right) - < I_D\left(t\right) >$, and $\delta I_A\left(t\right) = I_A\left(t\right) - < I_A\left(t\right) > \cdot < I_D(t) >$ and $< I_A\left(t\right) >$ are time average donor and acceptor intensities, respectively. Cross-correlation calculations were performed on the same traces used to generate the histograms and fit to a single exponential function, $y\left(x\right) = Ae^{\frac{-x}{\tau}} + y_0$.

## Structural representation of allosteric binding site by Chimera

Pairwise sequence alignment for PDB:7MTS, 7MTR, 7E9G, 7EPE, and 7EPF was performed using PDB:7MTS as the reference sequence. Alignment was based on best-aligning pair of chains and used the Needleman-Wunsch alignment algorithm. Unbound subunits and extracellular domains of mGluR2 were excluded prior to structure alignment. Specifically, residues L556-I816 (PDB: 7MTS, 7MTR, 7E9G) and G564-V825 (PDB:7EPE and 7EPF) were used for alignment. Allosteric pocket forming residues are from interacting residues in PDB:7MTS and previous mutagenesis studies (*Farinha et al., 2015*; *Seven et al., 2021*).

# Acknowledgements

We thank all members of the Reza Lab for thoughtful discussions and J Fei (University of Chicago) for providing MATLAB scripts. This work was supported by the National Institutes of Health grant R01GM140272 (to RV) and by The Searle Leadership Fund for the Life Sciences at Northwestern University and by the Chicago Biomedical Consortium with support from the Searle Funds at The Chicago Community Trust (to RV). BWL was supported in part by the National Institute of General Medical Sciences (NIGMS) Training Grant T32GM-008061.

# Additional information

## Competing interests

Hamid Samareh Afsari: is affiliated with Boehringer Ingelheim Pharma GmbH & Co. The author has no financial interests to declare. The other authors declare that no competing interests exist.

## Funding

| Funder | Grant reference number | Author |
|---|---|---|
| National Institute of General Medical Sciences | R01GM140272 | Reza Vafabakhsh |
| National Institute of General Medical Sciences | T32GM-008061 | Brandon Wey-Hung Liauw |
| National Institutes of Health | | Reza Vafabakhsh |
| National Institute of General Medical Sciences | | Brandon Wey-Hung Liauw |
| Chicago Community Trust | | Reza Vafabakhsh |

| Funder | Grant reference number | Author |
|---|---|---|
| Northwestern University | | Reza Vafabakhsh |
| Chicago Biomedical Consortium | | Reza Vafabakhsh |

The funders had no role in study design, data collection and interpretation, or the decision to submit the work for publication.

## Author contributions

Brandon Wey-Hung Liauw, Data curation, Formal analysis, Visualization, Writing – original draft, Writing – review and editing; Arash Foroutan, Data curation, Formal analysis, Writing – review and editing; Michael R Schamber, Writing – review and editing; Weifeng Lu, Formal analysis, Writing – review and editing; Hamid Samareh Afsari, Conceptualization, Data curation, Formal analysis, Supervision, Writing – review and editing; Reza Vafabakhsh, Conceptualization, Funding acquisition, Supervision, Writing – original draft, Writing – review and editing

### Author ORCIDs

Brandon Wey-Hung Liauw  http://orcid.org/0000-0002-6186-7092
Hamid Samareh Afsari  http://orcid.org/0000-0002-5839-4765
Reza Vafabakhsh  http://orcid.org/0000-0001-8384-3203

### Decision letter and Author response

Decision letter https://doi.org/10.7554/eLife.78982.sa1
Author response https://doi.org/10.7554/eLife.78982.sa2

# Additional files

### Supplementary files

- MDAR checklist

### Data availability

All data generated or analyzed during this study are included in the manuscript and supporting files. Accompanying source data is provided for figures 1-4 and tables 1-3. The PDB accession codes for human mGluR2 structures used are 7MTS, 7MTR, 7E9G, 7EPE, and 7EPF.

The following previously published datasets were used:

| Author(s) | Year | Dataset title | Dataset URL | Database and Identifier |
|---|---|---|---|---|
| Seven AB, Barros-Alvarez X, Skiniotis G | 2021 | CryoEM Structure of mGlu2 - Gi Complex | https://www.rcsb.org/structure/7MTS | RCSB Protein Data Bank, 7MTS |
| Seven AB, Barros-Alvarez X, Skiniotis G | 2021 | CryoEM Structure of Full-Length mGlu2 Bound to Ago-PAM ADX55164 and Glutamate | https://www.rcsb.org/structure/7MTR | RCSB Protein Data Bank, 7MTR |
| Lin S, Han S, Zhao Q, Wu B | 2021 | Cryo-EM structure of Gi-bound metabotropic glutamate receptor mGlu2 | https://www.rcsb.org/structure/7E9G | RCSB Protein Data Bank, 7E9G |
| Du J, Wang D, Lin S, Han S, Wu B, Zhao Q | 2021 | Crystal structure of mGlu2 bound to NAM563 | https://www.rcsb.org/structure/7EPE | RCSB Protein Data Bank, 7EPE |
| Du J, Wang D, Lin S, Han S, Wu B, Zhao Q | 2021 | Crystal structure of mGlu2 bound to NAM597 | https://www.rcsb.org/structure/7EPF | RCSB Protein Data Bank, 7EPF |

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
