## [Editor Report]

The authors advance our understanding of the molecular underpinnings of allostery in GPCRs by showing the effects of allosteric modulators of mGluR2 on receptor conformation at distinct sites in the presence and absence of orthosteric modulators. This is important as drugs and drug candidates acting outside the site where the orthosteric or endogenous ligands bind are harder to identify. This work provides insights into allosteric changes at the level of individual receptors and provides a new path for drug discovery that is of interest to studies of GPCRs in health and disease.

---

## [Decision Letter]

**Decision letter after peer review:**

Thank you for submitting your article "Conformational fingerprinting of allosteric modulators in metabotropic glutamate receptor 2" for consideration by *eLife*. Your article has been reviewed by 3 peer reviewers, including Marcel P Goldschen-Ohm as Reviewing Editor and Reviewer #1, and the evaluation has been overseen by Richard Aldrich as the Senior Editor. The following individuals involved in the review of your submission have agreed to reveal their identity: R. Scott Prosser (Reviewer #2); Terence E Hébert (Reviewer #3).

Essential revisions:

Please address the reviewers' comments below, which probably can be addressed by changes to the text.

*Reviewer #1 (Recommendations for the authors):*

Congratulations on a wonderful bit of work. The manuscript is excellent as is.

*Reviewer #2 (Recommendations for the authors):*

A few awkward grammar lines:

48 "and are important class of".

83 "at one stage of signalling cascade".

88 "to address this" as a start of a paragraph is poor style.

95 "for some canonical class A GPCR".

118 "of different orthosteric and allosteric ligands" needs a comma at the end.

277/278 vague.

395 "show that BINA can affect mGluR2 CRD conformation".

401 "still dynamic and receptor conformation".

462/463 "However, in the presence glutamate,".

543 "PAM stabilize the receptor dynamics".

*Reviewer #3 (Recommendations for the authors):*

A few suggestions – some comments on how labelling stoichiometry might or might not be an issue.

Next, although the correlation between efficacy as measured here is consistent with the literature, I always worry that modified receptors may have modified function (especially here for the new ECL2 construct). Was the function of these constructs tested directly?

Is it possible to examine FRET between distinct positions – i.e. VFT- CRD or CRD-ECL2 for example?

Finally, it would have been very interesting to see single-molecule FRET data for the ECL2 or VFT tagged receptors.

---

## [Author Response]

Essential revisions:Please address the reviewers' comments below, which probably can be addressed by changes to the text.

We thank the reviewers for careful reading of the manuscript and their comments and suggestions. We have revised the manuscript to improve clarity and expanded upon key discussion points based on reviewer suggestions. Changes are highlighted in the revised manuscript and explanation of revisions are detailed below. In addition, Hill slopes are added to Table 1 for live-cell FRET titration experiments and functional data for azi-CRD and azi-ECL2 sensors are now added to the supplemental figures (Figure 1 —figure supplement 3C-D).

Reviewer #1 (Recommendations for the authors):Congratulations on a wonderful bit of work. The manuscript is excellent as is.

We thank the reviewer for their kind words.

Reviewer #2 (Recommendations for the authors):A few awkward grammar lines:

We thank the reviewer for these suggestions and have revised the manuscript to improve its clarity. Specific changes are outlined below.

48 "and are important class of".

Revised to: “Factors that bind at non-cognate ligand binding sites to alter the allosteric activation process are classified as allosteric modulators and represent a promising class of therapeutics with distinct modes of binding and action.”

83 "at one stage of signalling cascade".

Revised to: "Generally, functional characterization of allosteric modulators is done using assays that quantify changes at specific steps of the signaling cascade, downstream of receptor, such as intracellular ca^2+^ levels, IP1 accumulation, cellular cAMP levels, ERK1/2 phosphorylation levels, or using energy transfer methods to quantify dissociation of signaling proteins."

88 "to address this" as a start of a paragraph is poor style.

Revised to: “Advances in methods for structure determination of membrane proteins have yielded atomic structures of many GPCRs bound to different allosteric modulators and provided insight into different ligand binding modalities and distinct modulator-induced conformations. However, despite these advances…"

95 "for some canonical class A GPCR".

Revised to: “in class A GPCRs”

118 "of different orthosteric and allosteric ligands" needs a comma at the end.

Corrected.

277/278 vague.

Revised to: “The ability of different mGluR2 PAMs to alter glutamate potency and efficacy as probed at each domain and to different degrees suggests that PAMs may utilize distinct mechanisms to achieve allosteric modulation of mGluR2, with each domain distinctly affected by each PAM.”

395 "show that BINA can affect mGluR2 CRD conformation".

Revised to: “Moreover, these single-molecule measurements demonstrated that the effect of BINA on mGluR2 conformation and dynamics depends on the presence or absence of glutamate.”

401 "still dynamic and receptor conformation".

Revised to: “Interestingly, even in the presence of 1 mM glutamate and BINA, the receptors remained dynamic with the CRDs not fully stabilized in a single conformation”

462/463 "However, in the presence glutamate,".

Revised to: “Interestingly, examination of the CRD dynamics by cross-correlation analysis revealed that the effect of MNI-137 on receptor dynamics is dependent on whether glutamate is present or not. In the absence of glutamate, MNI-137 reduced CRD dynamics (Figure 4 —figure supplement 1A). In contrast, when glutamate and MNI-137 were both present, we observed a glutamate concentration-dependent increase in the CRD dynamics (Figure 4F).”

543 "PAM stabilize the receptor dynamics".

Revised to: “On the other hand, in the presence of saturating agonist, the PAM reduced receptor dynamics while the NAM increased receptor dynamics (Figure 4 —figure supplement 2B).”

Reviewer #3 (Recommendations for the authors):A few suggestions – some comments on how labelling stoichiometry might or might not be an issue.Next, although the correlation between efficacy as measured here is consistent with the literature, I always worry that modified receptors may have modified function (especially here for the new ECL2 construct). Was the function of these constructs tested directly?

This is an important point. Indeed, we tested the functionality of the sensors. We have now included the glutamate dose-response curves for the azi-CRD and azi-ECL2 sensors in “Figure 1 —figure supplement 3” as panels C and D, respectively. Furthermore, we have revised the “Transfection and Protein Expression”, “Labeling for calcium imaging”, and “Calcium imaging” methods sections to include details about functional calcium imaging for azi-CRD and azi-ECL2.

Added to line 618: “For calcium imaging using unnatural amino acid containing proteins (azi-CRD or azi-ECL2), we followed the transfection and growth protocol described above and included an additional 1 μg of chimeric G protein (Gqo5).”

Added to line 649: “For cells expressing unnatural amino acid containing proteins, we labeled the cells with 4 µM Oregon Green 488 BAPTA-1.”

Changed line 689-692: “Sample was illuminated using a 488 nm laser and fluorescence from Oregon Green 488 was measured by a GaAsP-PMT detector with detection wavelengths set to 410-617 nm. For cells expressing SNAP-mGluR2 (no FLAG-tag), samples were excited using the 488 nm laser and a 640 nm laser simultaneously, and Cy5 fluorescence was measured with detection wavelengths set to 648-700 nm.”

Is it possible to examine FRET between distinct positions – i.e. VFT- CRD or CRD-ECL2 for example?

Examining FRET between different domains is possible in principle and could be interesting. Those experiments would answer a different question about intra-domain conformational change rather than inter-domain motion that we focused on in this manuscript. We anticipate that the development and validation of those sensors to be time consuming, due to the asymmetric nature of these sensors.

Finally, it would have been very interesting to see single-molecule FRET data for the ECL2 or VFT tagged receptors.

In our initial experiments we got similar conclusions when we did the experiments with the VFT domain sensor. Later, for the manuscript, we chose to invest the time towards a more thorough analysis of the CRD sensor as we believe it to be more representative of receptor function.

References:

Gregorio, G. G., Masureel, M., Hilger, D., Terry, D. S., Juette, M., Zhao, H., Zhou, Z., Perez-Aguilar, J. M., Hauge, M., Mathiasen, S., Javitch, J. A., Weinstein, H., Kobilka, B. K., and Blanchard, S. C. (2017). Single-molecule analysis of ligand efficacy in β(2)AR-G-protein activation. *Nature*, *547*(7661), 68-73. https://doi.org/10.1038/nature22354

Huang, S. K., Pandey, A., Tran, D. P., Villanueva, N. L., Kitao, A., Sunahara, R. K., Sljoka, A., and Prosser, R. S. (2021). Delineating the conformational landscape of the adenosine A(2A) receptor during G protein coupling. *Cell*, *184*(7), 1884-1894.e1814. https://doi.org/10.1016/j.cell.2021.02.041

Kaneko, S., Imai, S., Asao, N., Kofuku, Y., Ueda, T., and Shimada, I. (2022). Activation mechanism of the μ-opioid receptor by an allosteric modulator. *Proc Natl Acad Sci U S A*, *119*(16), e2121918119. https://doi.org/10.1073/pnas.2121918119

Levitz, J., Habrian, C., Bharill, S., Fu, Z., Vafabakhsh, R., and Isacoff, E. Y. (2016). Mechanism of Assembly and Cooperativity of Homomeric and Heteromeric Metabotropic Glutamate Receptors. *Neuron*, *92*(1), 143-159. https://doi.org/10.1016/j.neuron.2016.08.036

Maurel, D., Comps-Agrar, L., Brock, C., Rives, M. L., Bourrier, E., Ayoub, M. A., Bazin, H., Tinel, N., Durroux, T., Prézeau, L., Trinquet, E., and Pin, J. P. (2008). Cell-surface protein-protein interaction analysis with time-resolved FRET and snap-tag technologies: application to GPCR oligomerization. *Nat Methods*, *5*(6), 561-567. https://doi.org/10.1038/nmeth.1213

Nygaard, R., Zou, Y., Dror, R. O., Mildorf, T. J., Arlow, D. H., Manglik, A., Pan, A. C., Liu, C. W., Fung, J. J., Bokoch, M. P., Thian, F. S., Kobilka, T. S., Shaw, D. E., Mueller, L., Prosser, R. S., and Kobilka, B. K. (2013). The dynamic process of β(2)-adrenergic receptor activation. *Cell*, *152*(3), 532-542. https://doi.org/10.1016/j.cell.2013.01.008

Wang, X., Liu, D., Shen, L., Li, F., Li, Y., Yang, L., Xu, T., Tao, H., Yao, D., Wu, L., Hirata, K., Bohn, L. M., Makriyannis, A., Liu, X., Hua, T., Liu, Z. J., and Wang, J. (2021). A Genetically Encoded F-19 NMR Probe Reveals the Allosteric Modulation Mechanism of Cannabinoid Receptor 1. *J Am Chem Soc*, *143*(40), 16320-16325. https://doi.org/10.1021/jacs.1c06847

Wei, S., Thakur, N., Ray, A. P., Jin, B., Obeng, S., McCurdy, C. R., McMahon, L. R., Gutiérrez-de-Terán, H., Eddy, M. T., and Lamichhane, R. (2022). Slow conformational dynamics of the human A(2A) adenosine receptor are temporally ordered. *Structure*, *30*(3), 329-337.e325. https://doi.org/10.1016/j.str.2021.11.005

Wingler, L. M., Elgeti, M., Hilger, D., Latorraca, N. R., Lerch, M. T., Staus, D. P., Dror, R. O., Kobilka, B. K., Hubbell, W. L., and Lefkowitz, R. J. (2019). Angiotensin Analogs with Divergent Bias Stabilize Distinct Receptor Conformations. *Cell*, *176*(3), 468-478.e411. https://doi.org/10.1016/j.cell.2018.12.005